# Moderate Salinity of Nutrient Solution Improved the Nutritional Quality and Flavor of Hydroponic Chinese Chives (*Allium tuberosum* Rottler)

**DOI:** 10.3390/foods12010204

**Published:** 2023-01-03

**Authors:** Bojie Xie, Xuemei Xiao, Haiyan Li, Shouhui Wei, Ju Li, Yanqiang Gao, Jihua Yu

**Affiliations:** 1College of Horticulture, Gansu Agricultural University, Lanzhou 730070, China; 2State Key Laboratory of Aridland Crop Science, Lanzhou 730070, China; 3College of Water Conservancy and Hydropower Engineering, Gansu Agricultural University, Lanzhou 730070, China

**Keywords:** sodium chloride, nutritional quality, flavor, Chinese chive, volatile compounds

## Abstract

Sodium chloride (NaCl), as a eustressor, can trigger relevant pathways to cause plants to produce a series of metabolites, thus improving the quality of crops to a certain extent. However, there are few reports on the improvement of nutrient quality and flavor of hydroponic Chinese chives (*Allium tuberosum* Rottler) by sodium chloride. In this study, five NaCl concentrations were used to investigate the dose-dependent effects on growth, nutritional quality and flavor in Chinese chives. The results show that 10 mM NaCl had no significant effect on the growth of Chinese chives, but significantly decreased the nitrate content by 40% compared with 0 mM NaCl treatment, and the content of soluble protein and vitamin C was increased by 3.6% and 2.1%, respectively. In addition, a total of 75 volatile compounds were identified among five treatments using headspace solid-phase microextraction gas chromatography/mass spectrometry (HS-SPME/GC-MS). Compared with the 0 mM NaCl treatment, 10 mM NaCl had the greatest effect on the quantity and content of volatile compounds, with the total content increased by 27.8%. Furthermore, according to the odor activity values (OAVs) and odor description, there were 14 major aroma-active compounds (OAVs > 1) in Chinese chives. The “garlic and onion” odor was the strongest among the eight categories of aromas, and its highest value was observed in the 10 mM NaCl treatment (OAVs = 794).Taken together, adding 10 mM NaCl to the nutrient solution could improve the nutritional quality and flavor of Chinese chives without affecting their normal growth.

## 1. Introduction

Vegetables are one of the essential foods in people’s daily diets, providing a variety of nutrients for people, including sugar, minerals, vitamins, dietary fiber and so on [1,2]. In addition, the bioactive substances contained in different vegetables have special health care effects on the human body, such as lycopene in tomatoes, glucosinolates in cruciferous vegetables and prostaglandins in onions [3,4,5]. Therefore, the improvement of vegetable quality can enrich its nutritional and medicinal value, improve its market competitiveness and increase farmers’ income. There are many determinants of vegetable quality, including crop varieties, cultivation methods, management measures, environmental conditions and so forth [6,7]. Hence, we can take corresponding measures according to these influencing factors to obtain high-quality vegetables.

Chinese chive (*A. tuberosum* Rottl. ex. Spreng.) is a popular vegetable in Asia. It originated in China and is planted all over the country [8]. In addition to being a food, Chinese chive also has a certain medical value, because it contains rich active pharmaceutical ingredients, such as linalool, sulfides and flavonoid glycosides [9,10]. Moreover, the distinctive odor of Chinese chive is also one of the reasons for its popularity [11]. The distinctive odor of Chinese chive is mainly attributed to sulfur-containing compounds, which are produced by the degradation of S-alk(en)yl cysteine sulphoxide (CSO) [12]. In the field, the *Bradysiaodoriphaga* are attracted by this odor. Therefore, *Bradysiaodoriphaga* has become the main pest of Chinese chive, causing substantial reduction in Chinese chive production [13]. To solve this problem, farmers have applied large quantities of various pesticides to the roots of Chinese chives, which can reduce insect damage, but causes a high content of pesticide residues in Chinese chive leaves. It affects the safety of consumption and the sustainable development of the Chinese chive industry. The root systems of hydroponic vegetables are immersed in nutrient solution, and the anaerobic environment can prevent the occurrence of pests. Hence, hydroponic Chinese chive cultivation has become a new cultivation method of Chinese chives. According to the actual production effect, there is no *Bradysiaodoriphaga* in the hydroponic Chinese chive; however, it was also found that its quality is significantly lower than that of soil culture products in the same crop period [12]. For this reason, we need to take certain measures in the cultivation process to solve the problem of quality degradation of hydroponic Chinese chives.

During the growth and development of vegetable crops, appropriate environmental conditions are crucial for quality formation, including soil conditions, light, temperature, humidity, moisture, carbon dioxide (CO_2_) and nutrients [14,15]. For hydroponic vegetables, the precise management of the nutrient solution provides appropriate environmental conditions. Moderate salinity of nutrient solution as eustress (positive stress) can trigger relevant signal pathways to cause plants to produce a series of metabolites, thus improving the quality of vegetable crops, including nutritional quality, flavor, bioactive compounds and physical characteristics [16,17,18]. Research showed that melon firmness increased more with the application of saline water (6.1 dSm^−1^) than with control water (1.3 dSm^−1^) [19]. Adding 0 to 80 mM NaCl in the nutrient solution led to an increase in the fruit firmness of soilless melon [20]. Marín et al. [21] found that 30 mM NaCl increased the vitamin C content of red fruit in pepper. Cardeñosa et al. [22] reported that salinity stress promoted the accumulation of total phenols and antioxidant activity in strawberry fruits. In addition, the glucosinolate contents of *Brassicaceae* family vegetables can also be increased by salinity [23,24,25]. The flavor of a product is determined by its own quality attributes and the human olfactory system. The quality attributes include sourness, sweetness, bitterness, astringency and volatile compounds, whereas volatile compounds include aldehydes, ethers, alcohols, ketones, esters and so forth [26]. Previous research has shown that increasing salinity can improve the aroma and taste of many vegetables. Titratable acidity (TA) and total soluble solids (TSS) content were increased by adding inorganic salt to nutrient solution in cucumber fruits [27,28]. Similar results were found in cauliflower head [24], melon [20], strawberry [29,30] and watermelon [31]. However, there are few reports on the improvement of the nutritional quality and flavor quality of Chinese chives by moderate salinity. Our rationale is that moderate salinity in nutrient solution can trigger related metabolic pathways of Chinese chive quality formation and improve the content of metabolites, so as to improve its nutritional quality and flavor quality. We then further explain the reason for the improvement of flavor by using the odor activity value. Our findings could provide a theoretical basis for realizing high-quality production of hydroponic Chinese chives.

## 2. Materials and Methods

### 2.1. Plant Material and Experimental Design

The seeds of the Chinese chive cultivar “Jiuxing 18” were used as the experimental material for this study. On 10 December 2020, the seeds were sterilized with 1% copper sulfate solution and then sown in a 32-cell tray filled with a mixture of perlite and vermiculite (1:1, *v*:*v*). The seedings were irrigated with 1/4 strength Japan Chiba farming vegetable nutrient solution once a day, as described previously [12]. On 8 February 2021, the seedlings were transplanted into the hydroponic cultivation system of a modern glass greenhouse located in the Gansu Agricultural University of Lanzhou (36°30′ N, 103°40′ E). The temperature and relative humidity of the greenhouse were 20 ± 3 °C/15 ± 3 °C (day/night) and 60–70%, respectively. The pH of the solution was adjusted to 6.2 with H_3_PO_4_. The concentration of the solution was the total nutrient solution and was renewed every 5 days.

When the height of the seedings reached 30 cm (8 June 2021), we cut off the old leaves of the plant to grow new leaves. At the same time, the hydroponic Chinese chives were randomly divided into 5 groups, each group having 5 rectangular hydroponic boxes. The cover of hydroponic box had 11 holes, and 4 seedings were fixed in every hole. The salinity of the nutrient solution was regulated by adding NaCl to the nutrient solution to the following concentration: 0 mM, 5 mM, 10 mM, 20 mM and 30 mM. Other management of the nutrient solution was the same as the above description. On 8 August 2021, healthy Chinese chive leaves of the same size were selected for sampling from each treatment, and each treatment was repeated three times. All samples were ground into powder in liquid nitrogen and stored at −80 °C until analysis. The overall flowchart of the experiment and the hydroponic system are shown in Figure 1.

### 2.2. Determination of Growth Index and Photosynthetic Pigment Content of Chinese Chive

During harvest, 30 seedings were selected for each treatment to measure the plant height, leaf length and pseudo-stem diameter, and leaf width with a tape measure and vernier caliper, respectively. After harvest, the shoots and roots of the Chinese chives were dried at 105 °C for 30 min, and then at 75 °C to a constant weight for recording fresh weight and dry weight [32]. Root activity was determined using the triphenyl tetrazolium chloride (TTC) method [33] with slight modifications. Briefly, 0.5 g of fresh root tips were cut into 1 cm segments and placed in test tubes containing 5 mL of 1% TTC and 5 mL of 100 mM phosphate buffer (pH 7.5). After holding at 37 °C for 1h, the reaction was terminated by adding 2 mL of 1 M H_2_SO_4_. Subsequently, the roots were transferred to a mortar containing 3–5 mL of ethyl acetate (compared with methanol, ethyl acetate has low toxicity, so ethyl acetate was used instead) and a small amount of quartz sand. The grinding homogenate was filtered to the graduated test tube, and the residue was washed 2–3 times with a small amount of ethyl acetate. Finally, the volume of ethyl acetate was fixed at 10 mL, and then the absorbance values were performed with a UV-1780 spectrophotometer at 485 nm.

The photosynthetic pigment content was measured according to a previous study with some modifications [34]. Briefly, 0.1 g of fresh leaves were accurately weighed into the test tube, and then 10 mL of 80% acetone was added. The mixture was extracted in the dark for 48h until the leaves turned white(to ensure complete extraction) and shaken many times during this period (to ensure adequate contact between the leaves and acetone). After extraction, the optical density (OD) was measured with a UV-1780 spectrophotometer at 663 nm and 645 nm. The photosynthetic pigment content was calculated with the following formulas:Chl. a (mg∙g^−1^ FW) = (12.71 × OD_663_ − 2.59 × OD_645_) × V/FW(1)
Chl. b (mg∙g^−1^ FW) = (22.88 × OD_645_ − 4.67 × OD_663_) × V/FW(2)
Chl. T (mg∙g^−1^ FW) = (20.29 × OD_645_ + 8.04 × OD_663_) × V/FW(3)
where V and FW are the total volume of acetone and the fresh weight of the sample, respectively.

### 2.3. Determination of Nutritional Quality of Chinese Chive

The soluble sugar content was determined using the anthrone colorimetric method [35]. The soluble protein content was determined using the Coomassie brilliant blue method [36]. The vitamin C content was determined using the 2,6-dichloroindophenol stain method [37]. The nitrate content was determined according to the salicylic acid methods [38].

### 2.4. Determination of Volatile Flavor Compounds of Chinese Chive

The volatile flavor compounds of Chinese chive were determined using headspace solid-phase microextraction-gas chromatography/mass spectrometry (HS-SPME/GC-MS) [39]. Fresh Chinese chive leaves (1.5 g) were quickly ground into a homogenate and placed in a headspace vial fitted with the PTFE/silicone septa, and then ultrapure water (2 mL), Na_2_SO_4_ (0.75 g), 4 µL of difurfuryl sulfide (21.4 mg/L) and a magnetic stirring rotor were added into the vial. In order to extract more compounds, the headspace vial was heated at 70 °C for 15 min for equilibrating. After equilibration, the 85 µm CAR/PDMS fiber (Sigma-Aldrich, St. Louis, MO, USA) was inserted into the headspace vial for extracting with heating and agitation (50 min). At last, the SPME fiber was desorbed for 5 min in GC-MS (Agilent 7890B-7000D, Agilent, Santa Clara, CA, USA) using splitless mode.

The conditions of gas chromatography (GC) and mass spectrometry (MS) were as follows: DB-WAX capillary column (30 m × 0.25 mm, 0.25 µm); carrier gas and flow rate: He (>99.999% purity) at 1 mL/min; temperature program: initially 40 °C held for 1 min, increased to 80 °C at a rate of 8 °C/min, then increased to 130 °C at 2 °C/min and finally increased to 220 °C at 6 °C/min maintained for 3 min; MS ionization, electron ionization (EI), ionization energy: 70 eV; source temperature: 230 °C; scan area: 30–660 amu.

Matching score and retention index (RI) were used for the qualitative analysis of volatile compounds. For matching score, after comparison with the mass spectrometry library (NIST 2014), only compounds with a matching score of more than 70 were maintained. The RI was calculated on a DB-WAX chromatographic column with a C7–C40 n-alkanes series as external references under the same chromatographic conditions. The quantitative analysis of volatile compounds was performed according to the internal standard method, and the formula is as follows [40]:(4)$$\mathrm{Content}\;(µg/\mathrm{kg})=\frac{S1}{S2}\;\times \;\frac{M1}{M2}\;\times \;1000$$
where S1 and S2 represent the peak area of detected composition and the internal standard, respectively; M1 and M2 represent the amounts of the internal standard (µg) and the sample (g), respectively.

### 2.5. Calculation of Odor Activity Values

Odor activity values (OAVs) reflect the contribution of volatile compounds to the overall flavor. The calculation formula is as follows: OAVs = Ci/OTi(5) (Ci: the actual concentration of a certain volatile in Chinese chive; OTi: the odor threshold of the corresponding volatile) [41].

### 2.6. Statistical Analysis

The experimental data were analyzed using single factor variance (ANOVA) and Duncan’s multiple range tests of variance (*p* < 0.05) with SPSS 22.0 software (SPSS Inc., Chicago, IL, USA). All data were expressed as means ± standard error (SE). A Principal Components Analysis (PCA) was performed to determine differences between treatment groups. All figures were generated using OriginPro 2021 (OriginLab, Northampton, MA, USA).

## 3. Results

### 3.1. Effects of Adding Sodium Chloride in Nutrient Solution on Growth and Photosynthetic Pigment Content of Hydroponic Chinese Chive

As shown in Figure 2, the sodium chloride level had a significant effect on the growth and photosynthetic pigment content of the hydroponic Chinese chive. Compared with 0 mM treatment, 5–20 mM NaCl treatments had no significant effect on plant height and leaf length; however, 30 mM NaCl significantly decreased plant height and leaf length (Figure 2A). The pseudo-stem diameter treated with NaCl was significantly lower than the control (0 mM), especially 30 mM NaCl, which decreased by 38%. As the concentration of NaCl increased, the leaf width and leaf number increased at first and then decreased, reaching the maximum at 10 mM. When treated with 5–20 mM NaCl, the fresh weight of the root had no significant effect compared with the control (0 mM); however, the dry weight of the root treated with NaCl was significantly lower than the control (0 mM) (Figure 2B). The fresh and dry weights of the shoot had no significant effects when treated with 5 and 10 mM NaCl. Similarly, compared with 0 mM treatment, 5 and 10 mM NaCl had no significant effect on the content of Chl.a andChl.T, whereas the content of Chl.a treated with 20 and 30 mM NaCl significantly decreased by 21 and 25%, respectively (Figure 2C). In addition, 5–30 mM NaCl treatments had no significant effect on the Chl.b content. As the concentration of NaCl increased, the root activity significantly increased by 40, 42, 77 and 79%, respectively, compared with 0 mM treatment (Figure 2D).

### 3.2. Effects of Adding Sodium Chloride in Nutrient Solution on Soluble Sugar, Soluble Protein, Vitamin C and Nitrate Content of Hydroponic Chinese Chive

Compared with 0 mM treatment, 5 and 10 mM NaCl treatments had no significant effect on the soluble sugar content; however, 20 and 30 mM NaCl significantly decreased the soluble sugar content by 50 and 40%, respectively (Figure 3A). With the increase in NaCl concentration, the content of soluble protein and vitamin C presented an uptrend as a whole, and increased by 9.6 and 3.2% from 0 to 30 mM NaCl, respectively (Figure 3B,C). Moreover, when treated with 5–30 mM NaCl, the nitrate content significantly decreased by 21, 40, 39 and 44%, respectively (Figure 3D).

### 3.3. Effects of Adding Sodium Chloride in Nutrient Solution on Volatile Flavor Compounds of Hydroponic Chinese Chive

As shown in Table 1, a total of 75 volatile compounds were detected in hydroponic Chinese chive by using HS-SPME/GC-MS technology. These volatile compounds can be classified into nine categories according to their chemical structures, including 14 aldehydes, 19 ethers, 11 alcohols, 7 ketones, 6 hydrocarbons, 9 esters, 2 phenols, 2 furans, and 5 others. In terms of the amounts of volatile compounds, 65 compounds were detected in the 10 mM NaCl treatment, which was the highest among all treatments, followed by the 20 mM treatment with 59 compounds. Additionally, 58 compounds were found in both the 5 mM treatment and the 20 mM treatment. The control group contained the least volatile compounds, with only 48 compounds. From the content point of view, the total content of volatile compounds in the 10 mM NaCl treatment was the highest, reaching 18,733.52 μg/kg and increasing by 27.8% compared with the 0 mM treatment (14,659.02 μg/kg), followed by the 5 mM treatment with 16,408.35 μg/kg. However, 20 and 30 mM NaCl decreased the content of volatile compounds by 22.3 and 14.5%, respectively, compared with the 0 mM treatment.

There were 14 aldehydes detected in all treatment groups, of which 11 were detected in each treatment (Table 1). Compared with the 0 mM treatment, more aldehydes were detected in the NaCl treatment, such as (E)-2-octenal, 2-undecenal and (E)-2-tridecenal. Treatment with 20 and 30 mM NaCl induced production of all 14 aldehydes. The amounts of ethers were the largest among the volatile compounds of Chinese chive, followed by aldehydes. Treatment with 10 mM NaCl promoted the production of four ethers compared with 0 mM treatment, namely 2,4-dimethylthiophene, methyl propyl disulfide, methyl propyl trisulfide and 1-allyl-3-propyltrisulfane. Although the amounts of ethers treated with 5, 20 and 30 mM NaCl were the same, the constituent components were different; for example, methyl propyl trisulfide was detected in the 20 and 30 mM NaCl treatments, while 3-ethenyl-3,6-dihydrodithiine was only detected in the 5 mM treatment. Additionally, most kinds of ketones and esters in Chinese chive were obtained under 5 mM NaCl treatment, while the alcohols were lower than other NaCl treatments. For other compounds, 30 mM NaCl promoted the production of more compounds, such as 2-methyl-2-phenyl oxirane and dodecanoic acid.

As shown in Figure 4 and Table 1, the contents of various volatile compounds in different treatments of Chinese chive had great differences. Ethers (7465.47–13,585.64 μg/kg) were the most abundant of all compounds, followed by aldehydes (1065.56–2324.82 μg/kg). The phenols content (56.04–75.57 μg/kg) was the lowest, and furans (14.59–140.52 μg/kg) were only detected in the 5 and 10 mM treatment. Compared with 0 mM NaCl treatment, 5–30 mM NaCl significantly promoted the content of aldehydes, especially the 10 mM NaCl treatment, which increased by 118%. However, when the concentration of NaCl exceeded 10 mM, the content of ethers significantly decreased. The 10 mM NaCl treatment significantly promoted the content of alcohols and ketones, and the 5–20 mM NaCl treatment significantly promoted the content of hydrocarbons.

The Venn diagram shows common and specific volatile compounds among different treatments of hydroponic Chinese chive (Figure 5). A total of 41 volatile compounds were common substances among the five treatments. The 41 common compounds comprised 11 aldehydes, 13 ethers, 4 alcohols, 2 ketones, 1 hydrocarbon, 6 esters, 2 phenols and 2 other compounds. Treatment with 5, 10, 20 and 30 mM NaCl produced one, four, one and three specific compounds, respectively. The one specific compound produced by 5 and 20 mM NaCl treatment was 2,3-dihydrofuran and 7-tetradecyne, respectively. The four special compounds produced by 10 mM NaCl treatment were methyl propyl disulfide, (Z)-2-penten-1-ol, 1-hexadecanol and 2-pentylfuran, of which the content of 1-hexadecanol was the highest, reaching 45.54 μg/kg. Treatment with 30 mM NaCl produced three specific compounds: *cis*-4-Isopropenyl-1-methylcyclohexanol, 2,3,4-trimethyl-2-cyclopenten-1-one and 2-methyl-2-phenyl oxirane. Of these, *cis*-4-Isopropenyl-1-methylcyclohexanol was the most abundant, reaching 56.41 μg/kg.

### 3.4. Odor Activity Values Analysis and Radar Fingerprint Chart of Volatile Compounds in Chinese Chive

In general, volatile compounds with OAVs > 1 have an actual contribution to the overall flavor, that is, major aroma-active compounds. As shown in Table 2, there were only 14 volatile compounds with OAVs > 1, mainly aldehydes and ethers, but also ketones, phenols and furan. Moreover, the compounds with OAVs < 1 also have a certain effect on the overall flavor. According to the odor description, the aroma of volatile compounds in Chinese chive can be divided into eight categories, including “green and grassy”, “floral”, “fatty”, “sweet”, “garlic and onion”, “fresh”, “spicy” and “fruity” (Figure 6). The “garlic and onion” odor was the strongest scent, which mainly derived from ethers. Dimethyl disulfide, dimethyl trisulfide and diallyl disulfide are three major aroma-active compounds in ethers. Due to the lower threshold (6 μg/kg) and high concentration of dimethyl trisulfide, it contributed greatly to the “garlic and onion” odor. Among these treatments, the 10 mM NaCl treatment had a stronger “garlic and onion” odor than other treatments because it promoted the production of more ethers.

As shown in the radar map, the outline gradually changed after treatment with different concentrations of NaCl. The “fruity”, “floral” and “fatty” odors of treatment with 5 mM NaCl were stronger than in other treatments. The “fruity” odor mainly came from aldehydes, especially nonanal, whose threshold was only 1 μg/kg, so we could easily smell its aroma. Besides the “fruity” odor, nonanal also contained a “floral” and “fatty” odor. The other two major contributors to “floral” were decanal and β-ionone, both of which were major aroma-active compounds. The “fatty” odor was composed of aldehydes, containing (E)-2-octenal in addition to the five aldehydes that were common in the 10, 20 and 30 mM NaCl treatments. Compared with 0 mM treatment, NaCl treatments strengthened the “green and grassy” and “fresh” and “spicy” odor of Chinese chive, but the “sweet” odor was weakened when the NaCl concentration reached 20 mM. The “sweet” odor mainly came from the decanal mentioned above and two phenols of 2-methoxy-4-vinylphenol and eugenol, which also contributed to the “spicy” odor of Chinese chives. The “fresh” odor came from just two compounds, but only one of them was a major aroma-active compound, that is, (E)-2-octenal, whose content determined the intensity of “fresh” odor in Chinese chives. There were as many as 11 compounds in Chinese chive that could contribute to the “green and grassy” odor, and the two most important compounds were trans, trans-2,4-heptadienal and (2E,4E)-2,4-octadienal, which were the major aroma-active compounds.

### 3.5. Principal Component Analysis

The principal component analysis (PCA) of growth and quality parameters, as well as the 75 volatile compounds in this study, are shown in Figure 7. The sum of the first two principal components of growth and quality parameters reached 77.3%, of which PC1 and PC2 accounted for 53.6% and 23.7% of the total variance, respectively (Figure 7A). The first principal component sorted the treatments into two groups: 0, 5, and 10 mM were close to one group, while 20 and 30 mM were close to one group. It could be observed that the soluble sugar and Chl.b showed strong loadings with the first and second principal component, respectively. The root activity was highly negative to PC1, and nitrate was highly negative to PC2. The sum of the first two principal components of the 75 volatile compounds reached 57.3%, of which PC1 and PC2 accounted for 32.1% and 25.2% of the total variance, respectively (Figure 7B). The second principal component sorted the treatments into two groups: 0, 5, and 10 mM were close to one group, while 20 and 30 mM were close to one group, which was consistent with the results of growth and quality parameters.

## 4. Discussion

Adding different concentrations of sodium chloride in nutrient solution can induce different biochemical, morphological and physiological responses in plants. In the present study, we found that except for pseudo-stem diameter, low concentrations of NaCl treatment (5 and 10 mM) did not significantly inhibit the growth of Chinese chive and even had a certain extent of promotion (Figure 2A), which is consistent with the findings of Wang et al. [45] and Cheng et al. [46]. Wang et al. reported that the shoot growth of canola was promoted by 50 mM NaCl. Cheng et al. found that 10 mM NaCl promoted the growth of cassava by increasing and changing the morphological characteristics and number of root cells. To sum up, a low sodium chloride level of nutrient solution doesn’t inhibit plant growth and even has a certain growth-promoting effect on some plants. Dry and fresh weight is an important index reflecting the growth status of plants and the content of nutrients and structural substances. Our results show that low concentrations of NaCl (5 and 10 mM) had no significant effect on the fresh weight of the root or the fresh and dry weight of the shoot, while high concentrations of NaCl inhibited the fresh and dry weight of the root and shoot (Figure 2B). High concentration of NaCl caused salt stress on Chinese chives, resulting in reduced uptake of nutrients and water by roots, thus affecting the growth and development of plants [47,48]. Photosynthesis is the basis of plant growth and development, and chlorophyll is the key substance of photosynthesis. Therefore, chlorophyll content is an important indicator of plant growth. This experimental study showed that salinity mainly affected the content of Chl.a and Chl.T. When the concentration of NaCl reached 30 mM, the decrease in Chl.T was less than that in Chl.a due to no significant effects of salinity on Chl.b, but overall, the content of Chl.a and Chl.T decreased when the NaCl concentration reached 20 mM (Figure 2C), which was similar to the results of previous studies on alfalfa [49]. This may be due to a large amount of reactive oxygen species produced in Chinese chives under high concentrations of NaCl, which destroyed the structure of chloroplasts and slowed the synthesis of chlorophyll, resulting in a decrease in chlorophyll content [50]. The root is an important organ for plants to absorb water and nutrients, and its growth, development and root activity directly affects the life activities of individual plants. In this study, we found that with the increase in NaCl concentration, the root activity of Chinese chive was continuously enhanced, reaching the highest value at 30 mM (Figure 2D). The results were consistent with the findings of Wang et al. [51], who reported that 200 mM NaCl treatment significantly increased the root activity of *Limonium bicolor*, compared with 0 mM treatment. However, Zhang et al. [52] reported that 150 mM NaCl significantly decreased the root activity of wheat seedlings. This indicates that different plants at different growth stages respond differently to NaCl. Therefore, it is necessary to determine the optimum concentration of NaCl through corresponding pre-experiments in actual production.

With the improvement of people’s living standards, safe, healthy and high-quality agricultural products have become a mainstream demand in the agricultural market [16]. Therefore, how to improve the quality of agricultural products through corresponding agronomic measures has become one of the hotspots of current research. Compared with conventional breeding and genetic transformation, improving crop quality through environmental and agronomic factors has the advantages of safety, speed, high success rate and wide adaptability, and can also reduce consumers’ worries about genetically transformed or modified products [16,18]. At present, it has become a practicable method to improve the quality of crops by adding NaCl in nutrient solution [17]. This study found that the content of soluble sugar significantly decreased when the NaCl concentration reached 20 mM (Figure 3A), which may be due to the high concentration of NaCl-caused abiotic stress to Chinese chives, resulting in the decrease in chlorophyll content, then the weakening of photosynthesis, and finally, the reduction in soluble sugar content. However, compared with the 0 mM treatment, the content of soluble protein and vitamin C presented an uptrend with the increase in NaCl concentration (Figure 3B,C), which was similar to the results of Abdullah et al. [53] and Li et al. [54]. Soluble protein is an important osmotic regulator, so increasing it under salt stress helps to improve the stress resistance of plants and also improves the nutritional quality to a certain extent. In addition, we observed that NaCl treatments significantly decreased the nitrate content compared with the control (0 mM) (Figure 3D), which may be because related genes encoding nitrate reductase enzyme were reduced by NaCl [55]. In general, this study demonstrated that adding NaCl in a nutrient solution improved the nutritional quality and safety quality of Chinese chives.

Compared with the nutritional value of Chinese chives, it seems that people are more familiar with their unique flavor. The main source of the unique aroma of Chinese chive is secondary metabolites, such as sulfur-containing compounds [56,57]. Salinity can induce plants to produce related secondary metabolites, so it can be inferred that moderate salinity can improve the flavor quality of Chinese chives. In this study, we detected a total of 75 volatile compounds among the five treatments, which mainly comprised aldehydes, ethers, alcohols, ketones, hydrocarbons, esters, phenols, furans and other compounds (Table 1). Compared with 0 mM treatment, 5 and 10 mM NaCl treatments improved the quantity and content of volatile compounds, suggesting that moderate NaCl concentration stimulated the metabolic pathway related to the production of volatile compounds in Chinese chive, thereby improving its flavor quality. Similar results were found by Neffati [58], who reported that the essential oil yield of coriander leaves increased significantly under moderate salinity (25 and 50 mM NaCl) and decreased significantly under 75 mM NaCl. Chatterjee et al. [59] also found that 25 and 50 mM NaCl increased the green leaf volatiles of rice. Ethers are the main volatile compounds of *Allium* vegetables, which are called thioethers. Among the volatile compounds of Chinese chive, we detected 19 thioethers in five treatments with the largest quantity and content of all the compounds, which is consistent with previous research results [56,57]. Compared with 0 mM treatment, 5 and 10 mM NaCl increased the thioethers content by 3 and 13.8%, respectively (Figure 4), indicating that moderate NaCl concentration may stimulate the expression of key genes involved in the metabolic pathway of thioethers generation [12]. The content of aldehydes in the volatile compounds of Chinese chive was second only to that of ethers, but unlike ethers, the total content was increased in all NaCl treatment groups, indicating that the effects of nutrient solution salt on various substances were different (Figure 4). The content of trans-2-hexenal was the highest in aldehydes, which are released when plant cells are damaged under external stress [60]. Therefore, the growth status of plants can be judged by measuring the content of trans-2-hexenal in the air in the field, so as to take corresponding protective measures [61]. More alcohols were detected when the concentration of NaCl reached 10 mM, such as linalool, isophytol, 3-hexen-1-ol, etc. Linalool is a volatile compound that mainly exists in flowers. It not only endows plants with floral fragrance, but also participates in multiple biological processes of plant growth and development, such as pollinator attraction and plant defense [62]. In this experiment, only two phenols were detected in Chinese chive, namely eugenol and 2-methoxy-4-vinylphenol (Table 1), among which eugenol is not only the aromatic compound of many fruits [63,64], but also has antioxidant and antimicrobial properties [65]. Therefore, phenols and sulfur-containing compounds contribute to both the antioxidant and antimicrobial properties of Chinese chives.

When volatile compounds enter the human olfactory system, they react with olfactory receptors and eventually produce the corresponding olfaction in the cerebral cortex. The intensity of odor depends on two factors, the actual concentration and its odor threshold, of which the odor threshold refers to the lowest concentration that can be smelled by the human body [66,67]. The ratio between actual concentration and odor threshold is odor activity values (OAVs), which reflect the contribution of volatile compounds to the overall flavor [68]. In this study, we detected 14 major aroma-active compounds according to OAVs > 1 which are essential for the aroma quality of Chinese chive (Table 2). In addition, the volatile compounds with OAVs < 1 influence the flavor of the Chinese chive through internal synergistic effects [69]. Therefore, according to the odor description and OAVs of each volatile compound, we have drawn the radar fingerprints of volatile compounds in Chinese chives under different NaCl concentrations of nutrient solution (Figure 6). As shown in the radar map, the OAVs of the “garlic and onion” odor are much higher than others; that is why the aroma of Chinese chives is similar to garlic and onion. At the same time, the “garlic and onion” odor comes from thioethers, so it is crucial to improve the flavor quality of Chinese chive by increasing the content of thioethers, especially dimethyl trisulfide, which widely exists in *Allium* [70]. The “floral” odor was mainly comprised of nonanal and β-ionone, wherein β- Ionone is widely found in vegetables and fruits and is the major aroma-active compound in some plants [71,72]. The OAVs of the “fresh” and “spicy” odors were the lowest of the eight categories because there were few kinds and low concentrations of odor contributing compounds. Among the five treatments, the effects of NaCl treatments on the eight categories of odor were different, indicating that NaCl, as a eustressor, had different stimulating effects on the generation pathways of various volatile compounds in Chinese chive, which needs further research.

Principal component analysis (PCA) is a commonly used multivariate analysis method which can clearly show the differences between samples [73]. As shown in Figure 7, with the increase in NaCl concentration, each treatment was divided into different groups according to growth and quality parameters and 75 volatile compounds. For growth and quality parameters, 0, 5 and 10 mM were close to one group, indicating that low salinity would not inhibit the growth of Chinese chives, which was similar to the research results of Zhao et al. [74]. For volatile compounds, grouping was similar to growth and quality parameters, indicating that high salinity weakened the flavor of Chinese chives. Similar results were found in *Lycopersicum esculentum* leaves [75] and rosemary [76]. On the whole, adding 10 mM NaCl to the nutrient solution can improve the nutritional quality and flavor of Chinese chive without affecting their normal growth, which provides a theoretical basis for the high-quality cultivation of Chinese chive in hydroponics.

## 5. Conclusions

In this study, nutrient solution treatment with 10 mM NaCl had no significant effect on growth parameters, dry and fresh weight, photosynthetic pigment content and soluble sugar content of Chinese chive. Differently, treatment with 10 mM NaCl significantly decreased the nitrate content and increased the root activity. Moreover, soluble protein and vitamin C content were increased under all NaCl treatments. A total of 75 volatile compounds were detected in Chinese chive, including 14 aldehydes, 19 ethers, 11 alcohols, 7 ketones, 6 hydrocarbons, 9 esters, 2 phenols, 2 furans and 5 others. Treatment with 10 mM NaCl had the greatest effect on the quantity and content of volatile compounds, especially aldehydes and ethers. According to the odor activity values (OAVs), there are 14 major aroma-active compounds (OAVs > 1) in Chinese chives. The “garlic and onion” odor was the strongest among the eight categories of aroma in Chinese chives. The 10 mM NaCl treatment had the strongest “garlic and onion” odor among the different treatments. In conclusion, nutrient solution treatment with 10 mM NaCl could improve the nutritional quality and flavor of Chinese chives without affecting their normal growth. The research results provide a feasible application basis for improving the nutritional quality and flavor of the hydroponic Chinese chives. In addition, vegetables with stronger flavor or special medical value can be produced by using similar methods, which can not only meet people’s daily health care needs, but also increase farmers’ income and improve the local economic level.

## Figures and Tables

**Figure 1 foods-12-00204-f001:**
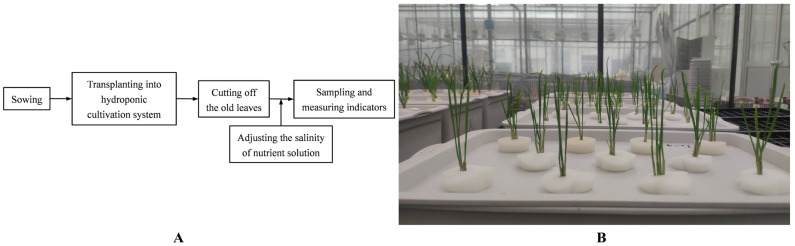
The overall flowchart of the experiment (**A**) and the hydroponic system (**B**). Each hydroponic box cover had 11 holes, and 4 seedings were fixed in every hole.

**Figure 2 foods-12-00204-f002:**
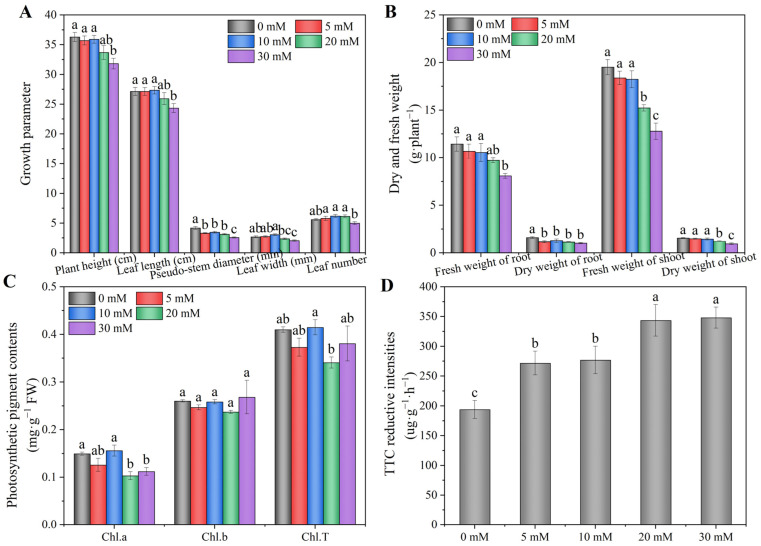
Effects of adding sodium chloride in nutrient solution on growth (**A**,**B**), photosynthetic pigment content (**C**) and root activity (**D**) of hydroponic Chinese chive. Values are represented as mean ± SE (*n* = 3). Different lowercase letters denote significant differences by Duncan’s multiple range test (*p* < 0.05).

**Figure 3 foods-12-00204-f003:**
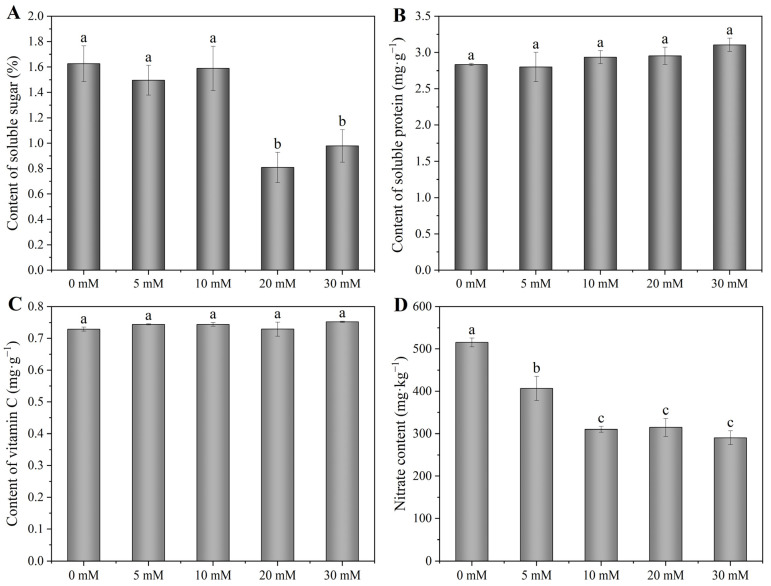
Effects of adding sodium chloride in nutrient solution on soluble sugar (**A**), soluble protein (**B**), vitamin C (**C**) and nitrate content (**D**) of hydroponic Chinese chive. Values are represented as mean ± SE (*n* = 3). Different lowercase letters denote significant differences by Duncan’s multiple range test (*p* < 0.05).

**Figure 4 foods-12-00204-f004:**
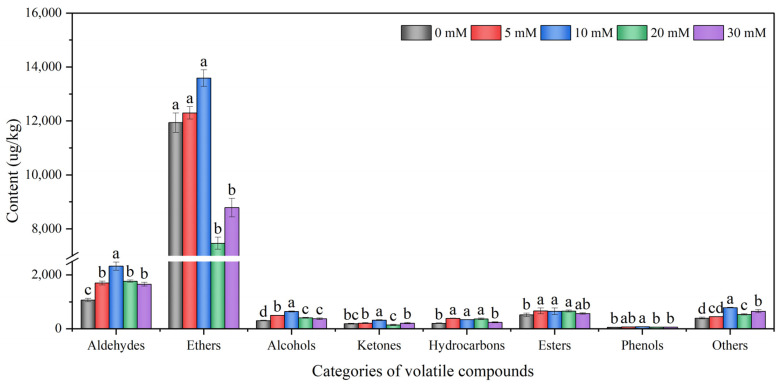
Effects of adding sodium chloride in nutrient solution on the content of volatile compounds of hydroponic Chinese chive. Values are represented as mean ± SE (*n* = 3). Different lowercase letters denote significant differences by Duncan’s multiple range test (*p* < 0.05).

**Figure 5 foods-12-00204-f005:**
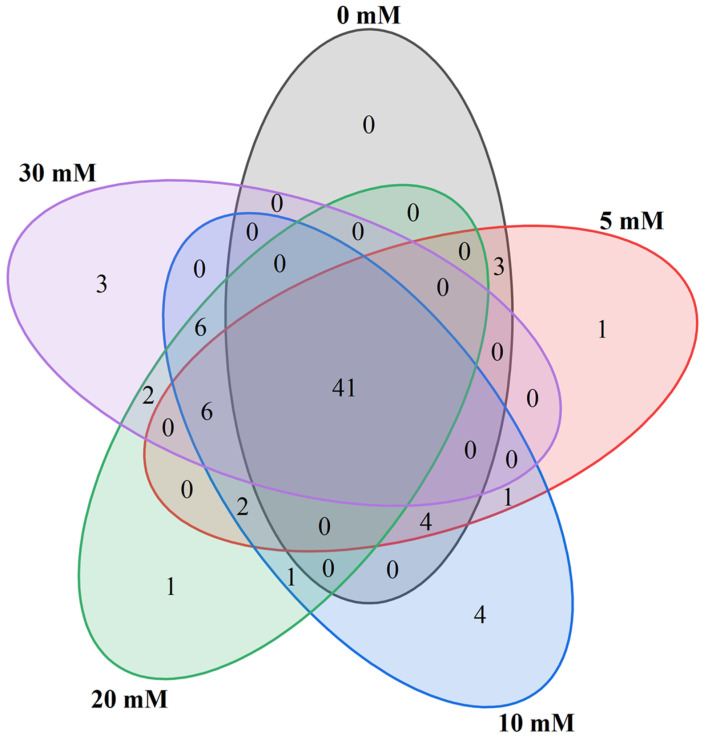
Effects of adding sodium chloride in nutrient solution on common and specific volatile compounds of hydroponic Chinese chive.

**Figure 6 foods-12-00204-f006:**
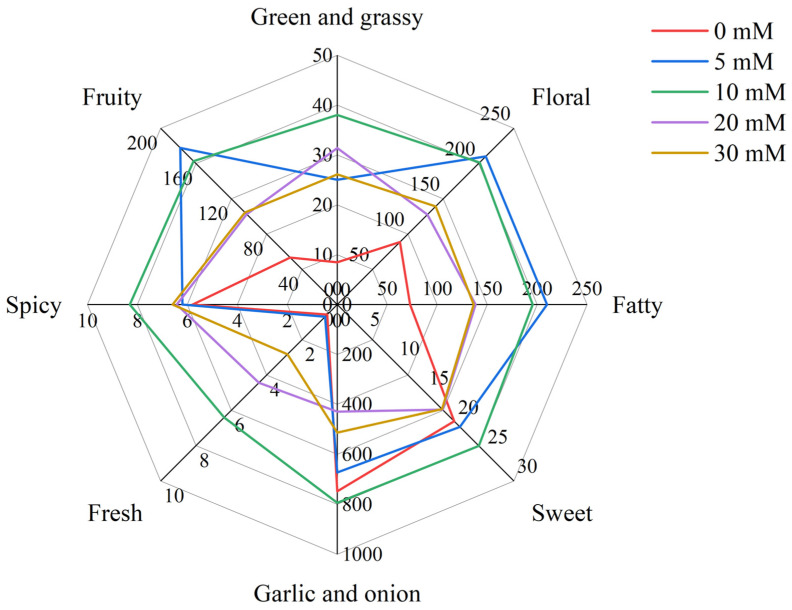
Radar fingerprint chart of volatile compounds in Chinese chive at different sodium chloride concentrations of nutrient solution.

**Figure 7 foods-12-00204-f007:**
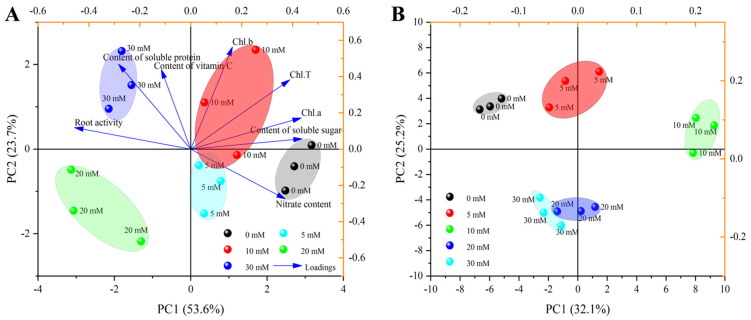
Principal component analysis (PCA) of growth and quality parameters (**A**) and 75 volatile compounds (**B**).

**Table 1 foods-12-00204-t001:** Effects of adding sodium chloride in nutrient solution on volatile compounds of hydroponic Chinese chive.

NO.	Volatile Compound	Formula	Content (μg/kg)					CAS
			0 mM	5 mM	10 mM	20 mM	30 mM	
	Aldehydes							
1	2-Butenal	C_4_H_6_O	155.96 ± 24.62ab	168.13 ± 19.87ab	227.31 ± 26.57a	77.09 ± 24.95c	127.82 ± 20.32bc	4170-30-3
2	*Trans*-2-Hexenal	C_6_H_10_O	318.29 ± 19.37a	373.42 ± 43.43a	519.25 ± 64.09a	461.28 ± 30.10a	398.62 ± 5.32a	6728-26-3
3	2-Ethyl-2-hexanal	C_8_H_14_O	10.83 ± 4.35a	17.16 ± 8.97a	26.56 ± 4.40a	19.52 ± 2.53a	20.22 ± 7.85a	645-62-5
4	Nonanal	C_9_H_18_O	52.28 ± 9.15b	175.15 ± 5.10a	150.55 ± 46.83a	97.56 ± 16.21ab	101.26 ± 17.24ab	124-19-6
5	2,4-Hexadienal	C_6_H_8_O	38.90 ± 4.97b	75.23 ± 9.83a	92.58 ± 1.36a	52.46 ± 6.15b	52.83 ± 7.39b	142-83-6
6	(E)-2-Octenal	C_8_H_14_O	—	—	19.21 ± 4.35a	13.30 ± 1.08ab	8.45 ± 1.46b	2548-87-0
7	Cyclopentanecarboxaldehyde, 2-methyl-3-methylene-	C_8_H_12_O	196.94 ± 7.89b	384.14 ± 17.64a	447.81 ± 8.40a	238.52 ± 39.68b	215.88 ± 12.91b	1000154-24-0
8	Decanal	C_10_H_20_O	26.80 ± 4.68a	23.75 ± 7.37a	23.08 ± 4.20a	21.16 ± 0.69a	20.60 ± 9.00a	112-31-2
9	*trans*,*trans*-2,4-Heptadienal	C_7_H_10_O	69.97 ± 20.73a	98.56 ± 3.34a	96.47 ± 10.46a	92.36 ± 5.50a	109.56 ± 35.75a	4313-03-5
10	(2E,4E)-2,4-Octadienal	C_8_H_12_O	23.85 ± 4.22b	161.00 ± 18.98a	203.45 ± 44.34a	193.71 ± 42.47a	146.78 ± 3.19a	30361-28-5
11	4-Ethylbenzaldehyde	C_9_H_10_O	107.60 ± 11.58c	135.47 ± 6.24bc	441.44 ± 92.57a	335.33 ± 51.21a	295.03 ± 62.63ab	4748-78-1
12	2-Undecenal	C_11_H_20_O	—	25.62 ± 5.85a	16.81 ± 1.96ab	13.46 ± 1.49b	10.51 ± 1.05b	2463-77-6
13	α-Methyl-α-vinyl-2-furanacetaldehyde	C_9_H_10_O_2_	64.15 ± 2.33a	58.03 ± 12.31a	60.30 ± 11.42a	59.22 ± 12.71a	49.13 ± 11.66a	31776-28-0
14	(E)-2-Tridecenal	C_13_H_24_O	—	—	—	86.46 ± 2.01b	93.60 ± 1.96a	7069-41-2
	Ethers							
15	Dimethyl disulfide	C_2_H_6_S_2_	629.51 ± 41.27bc	804.02 ± 101.09ab	991.61 ± 119.63a	417.95 ± 31.59c	541.75 ± 25.18c	624-92-0
16	2,4-Dimethylthiophene	C_6_H_8_S	—	70.86 ± 20.54ab	98.14 ± 13.41a	50.63 ± 9.04b	40.93 ± 7.29b	638-00-6
17	Methyl propyl disulfide	C_4_H_10_S_2_	—	—	18.71	—	—	2179-60-4
18	3,4-Dimethyl-thiophene	C_6_H_8_S	442.58 ± 48.32a	302.12 ± 43.36b	365.80 ± 38.22ab	268.87 ± 30.06b	287.81 ± 13.12b	632-15-5
19	Allyl methyl disulfide	C_4_H_8_S_2_	892.68 ± 71.45abc	1051.11 ± 141.06ab	1212.82 ± 145.39a	685.61 ± 55.11c	772.49 ± 34.71bc	2179-58-0
20	Methyldithio-1-propene	C_4_H_8_S_2_	2320.21 ± 148.07a	1996.12 ± 233.17a	1905.04 ± 107.89ab	1260.75 ± 64.79c	1514.93 ± 96.38bc	23838-19-9
21	Dimethyl trisulfide	C_2_H_6_S_3_	4102.49 ± 260.73a	3504.61 ± 463.87ab	4100.35 ± 493.11a	2239.13 ± 251.07c	2722.21 ± 200.21bc	3658-80-8
22	1-[[(Z)-prop-1-enyl]Disulfanyl]propane	C_6_H_12_S_2_	46.88 ± 5.58a	66.33 ± 3.58a	23.64 ± 5.95a	38.25 ± 2.39a	31.61 ± 8.98a	23838-20-2
23	(Z)-1-Allyl-2-(prop-1-en-1-yl)disulfane	C_6_H_10_S_2_	1049.52 ± 75.22b	1392.22 ± 94.07 ab	1621.96 ± 225.23a	498.90 ± 54.09c	920.84 ± 131.91bc	122156-03-0
24	Diallyl disulfide	C_6_H_10_S_2_	358.16 ± 52.04c	646.34 ± 92.57ab	835.60 ± 40.75a	642.65 ± 51.09ab	436.97 ± 86.68bc	2179-57-9
25	Methyl propyl trisulfide	C_4_H_10_S_3_	—	—	112.50 ± 19.90a	55.37 ± 7.89b	34.65 ± 7.49b	17619-36-2
26	3H-1,2-Dithiole	C_3_H_4_S_2_	317.14 ± 39.16a	467.26 ± 78.04a	463.22 ± 58.76a	276.42 ± 58.24a	330.99 ± 52.07a	288-26-6
27	Methyl allyl trisulfide	C_4_H_8_S_3_	1193.55 ± 33.08a	1317.75 ± 55.43a	1190.06 ± 132.13a	704.72 ± 70.42b	781.09 ± 148.20b	34135-85-8
28	(E)-1-Methyl-3-(prop-1-en-1-yl)trisulfane	C_4_H_8_S_3_	185.19 ± 61.48a	116.99 ± 16.47a	—	—	—	23838-25-7
29	1-Allyl-3-propyltrisulfane	C_6_H_12_S_3_	—	—	5.29 ± 0.64b	7.58 ± 1.02b	14.68 ± 1.56a	33922-73-5
30	3-Ethenyl-3,6-dihydrodithiine	C_6_H_8_S_2_	35.30 ± 2.01a	35.48 ± 11.57a	31.09 ± 0.87a	—	—	62488-52-2
31	3-Methyl-3H-1,2-dithiole	C_4_H_6_S_2_	184.04 ± 30.66ab	250.25 ± 55.25a	239.88 ± 39.35ab	121.42 ± 23.82b	142.57 ± 21.44ab	118023-96-4
32	Diallyl trisulfide	C_6_H_10_S_3_	126.01 ± 15.25a	236.69 ± 96.47a	364.79 ± 119.38a	191.00 ± 49.85a	207.33 ± 33.41a	2050-87-5
33	(Z)-1-Allyl-3-(prop-1-en-1-yl) trisulfane	C_6_H_10_S_3_	52.50 ± 3.90a	39.12 ± 12.48a	5.12 ± 1.11b	6.20 ± 1.17b	4.72 ± 0.20b	382161-75-3
	Alcohols							
34	(z)-2-Penten-1-ol	C_5_H_10_O	—	—	6.35	—	—	1576-95-0
35	3-Hexen-1-ol	C_6_H_12_O	—	—	15.69 ± 0.60a	9.71 ± 0.54b	9.38 ± 0.69b	544-12-7
36	Linalool	C_10_H_18_O	—	—	11.84 ± 7.03a	8.73 ± 0.92ab	6.49 ± 1.65ab	78-70-6
37	1-Octanol	C_8_H_18_O	6.59 ± 0.64b	23.12 ± 3.74a	21.58 ± 5.66a	14.15 ± 2.89ab	12.55 ± 0.87ab	111-87-5
38	(+)-Borneol	C_10_H_18_O	17.49 ± 1.11c	24.09 ± 1.28b	40.71 ± 2.32a	34.91 ± 1.83a	22.83 ± 2.46bc	464-43-7
39	*Cis*-4-Isopropenyl-1-methylcyclohexanol	C_10_H_18_O	—	—	—	—	56.41	7299-41-4
40	*Cis*-4-(isopropyl)-1-methylcyclohex-2-en-1-ol	C_10_H_18_O	54.99 ± 4.03b	90.10 ± 0.85a	53.33 ± 9.41b	—	—	29803-82-5
41	3,7,11,15-Tetramethyl-2-hexadecen-1-ol	C_20_H_40_O	81.42 ± 2.21c	131.38 ± 22.97bc	241.13 ± 19.37a	172.12 ± 17.89ab	138.26 ± 38.74bc	102608-53-7
42	Isophytol	C_20_H_40_O	—	—	23.65 ± 1.04a	10.34 ± 0.47b	—	505-32-8
43	1-Hexadecanol	C_16_H_34_O	—	—	45.54	—	—	36653-82-4
44	Phytol	C_20_H_40_O	140.88 ± 30.06a	232.30 ± 20.95a	181.01 ± 72.98a	154.47 ± 34.93a	123.66 ± 44.61a	150-86-7
	Ketones							
45	2,5-Octanedione	C_8_H_14_O_2_	—	10.90 ± 0.19b	15.46 ± 0.55a	8.39 ± 0.23c	—	3214-41-3
46	1-(1,4-dimethyl-3-cyclohexen-1-yl)-Ethanone	C_10_H_16_O	—	7.23 ± 1.23a	7.89 ± 1.15a	8.08 ± 0.98a	4.02 ± 0.49b	43219-68-7
47	2-Cyclopenten-1-one, 2,3,4-trimethyl-	C_8_H_12_O	—	—	—	—	12.48	28790-86-5
48	α-Ionone	C_13_H_20_O	—	15.72 ± 0.70b	28.01 ± 1.81a	—	—	127-41-3
49	Geranylacetone	C_13_H_22_O	33.82 ± 3.90a	41.59 ± 7.23a	—	—	—	3796-70-1
50	β-ionone	C_13_H_20_O	138.47 ± 16.68ab	120.68 ± 15.50ab	233.75 ± 62.46a	104.78 ± 26.57b	174.56 ± 27.75ab	79-77-6
51	Hexahydrofarnesyl acetone	C_18_H_36_O	13.88 ± 1.42b	13.71 ± 1.38b	34.47 ± 6.45a	18.99 ± 4.89b	17.93 ± 0.93b	502-69-2
	Hydrocarbons							
52	3-Ethyl-2-methyl-1,3-hexadien	C_9_H_16_	—	44.96 ± 15.12a	34.60 ± 3.33a	36.76 ± 0.70a	24.74 ± 4.07a	61142-36-7
53	(E)-7-Tetradecene	C_14_H_28_	—	—	—	11.14 ± 0.35a	8.82 ± 1.79a	41446-63-3
54	2-Tridecyne	C_13_H_24_	8.66 ± 1.05a	8.98 ± 2.11a	11.37 ± 1.18a	—	—	28467-75-6
55	7-Tetradecyne	C_14_H_26_	—	—	—	16.44	—	35216-11-6
56	Neophytadiene	C_20_H_38_	163.00 ± 59.96b	302.66 ± 20.15a	278.87 ± 1.11a	298.92 ± 29.97a	204.04 ± 18.65ab	504-96-1
57	3,4′-Diethyl-1,1′-biphenyl	C_16_H_18_	31.89 ± 9.77a	27.35 ± 5.26ab	13.23 ± 0.12bc	—	—	61141-66-0
	Esters							
58	*Trans*-3-Hexenyl acetate	C_8_H_14_O_2_	—	14.18 ± 1.55b	33.71 ± 3.69a	28.51 ± 2.49a	15.85 ± 1.19b	3681-82-1
59	Linalyl acetate	C_12_H_20_O_2_	6.39 ± 0.51b	9.17 ± 0.90a	—	—	—	115-95-7
60	Ethyl myristate	C_16_H_32_O_2_	41.07 ± 9.42a	46.05 ± 2.57a	31.54 ± 1.54a	33.20 ± 5.00a	21.65 ± 4.77a	124-06-1
61	Hexadecanoic acid, ethyl ester	C_18_H_36_O_2_	253.26 ± 49.58a	219.13 ± 41.37a	112.09 ± 40.42a	217.44 ± 9.81a	192.64 ± 53.67a	628-97-7
62	Dimethyl phthalate	C_10_H_10_O_4_	41.44 ± 4.86b	108.11 ± 23.62a	127.86 ± 38.11a	96.39 ± 18.48a	106.92 ± 31.18a	131-11-3
63	Ethyl 9-hexadecenoate	C_18_H_34_O_2_	20.45 ± 9.61a	24.29 ± 3.44a	26.85 ± 1.93a	16.76 ± 6.84a	16.59 ± 2.62a	54546-22-4
64	Linoleic acid ethyl ester	C_20_H_36_O_2_	—	21.90 ± 0.94a	22.04 ± 4.69a	19.12 ± 4.27a	—	544-35-4
65	1,2-Benzenedicarboxylic acid, bis(2-methylpropyl) ester	C_16_H_22_O_4_	133.37 ± 17.28a	182.43 ± 6.25a	250.05 ± 70.40a	221.87 ± 29.52a	178.57 ± 33.54a	84-69-5
66	9,12,15-Octadecatrienoic acid, ethyl ester, (Z,Z,Z)-	C_20_H_34_O_2_	19.09 ± 2.98c	36.84 ± 9.67ab	45.38 ± 2.57a	22.99 ± 3.68bc	35.14 ± 3.70abc	1191-41-9
	Phenols							
67	Eugenol	C_10_H_12_O_2_	19.51 ± 2.11bc	10.69 ± 2.05c	34.94 ± 4.27a	23.71 ± 5.54ab	25.39 ± 4.46ab	97-53-0
68	2-Methoxy-4-vinylphenol	C_9_H_10_O_2_	36.53 ± 6.48a	56.33 ± 8.77a	40.63 ± 6.03a	37.04 ± 12.52a	36.21 ± 9.01a	7786-61-0
	Furans							
69	2,3-Dihydrofuran	C_4_H_6_O	—	140.52	—	—	—	1191-99-7
70	2-Pentylfuran	C_9_H_14_O	—	—	14.59	—	—	3777-69-3
	Others							
71	2-Methyl-2-phenyl oxirane	C_9_H_10_O	—	—	—	—	2.37	2085-88-3
72	3-Ethoxy-3,7-dimethyl-1,6-octadiene	C_12_H_22_O	19.78 ± 0.75b	23.23 ± 1.98b	28.66 ± 0.75a	14.27 ± 1.00c	8.97 ± 1.91d	72845-33-1
73	Methoxyphenyloxim	C_8_H_9_NO_2_	375.69 ± 72.97c	423.07 ± 1.21bc	700.96 ± 64.18a	468.03 ± 29.05bc	573.06 ± 60.84ab	1000222-86-6
74	*trans*-Z-α-Bisabolene epoxide	C_15_H_24_O	—	4.72 ± 0.69cd	12.85 ± 3.56ab	15.78 ± 1.44a	7.23 ± 1.85bc	—
75	Dodecanoic acid	C_12_H_24_O_2_	—	—	42.45 ± 11.25b	36.20 ± 0.84b	61.37 ± 4.25a	143-07-7
	Total content		14,659.02	16,408.35	18,733.52	11,386.18	12,534.01	
	Total number		48	58	65	59	58	

“—”, not found. Values are represented as mean ± SE (*n* = 3). Different lowercase letters denote significant differences by Duncan’s multiple range test (*p* < 0.05).

**Table 2 foods-12-00204-t002:** Odor activity values (OAVs) and odor description of volatile compounds in Chinese chive at different sodium chloride concentrations of nutrient solution.

NO. ^a^	Volatile Compound	Odor Threshold ^b^ (μg/kg)	Odor Activity Values (OAVs)					Odor Description ^c^
			0 mM	5 mM	10 mM	20 mM	30 mM	
	Aldehydes							
1	2-Butenal	1400.0	0.11	0.12	0.16	0.06	0.09	Flower
2	Trans-2-Hexenal	1125.0	0.28	0.33	0.46	0.41	0.35	Green, banana, fatty
4	Nonanal	1.0	52.28	175.15	150.55	97.56	101.26	Fatty, orange, rose odor
5	2,4-Hexadienal	60.0	0.65	1.25	1.54	0.87	0.88	Sweet, green aroma
6	(E)-2-Octenal	3.0	—	—	6.40	4.43	2.82	Fresh, cucumber, fatty, green, herbal, banana
8	Decanal	2.0	13.40	11.88	11.54	10.58	10.30	Sweet, floral, fatty odor
9	*trans*,*trans*-2,4-Heptadienal	15.4	4.54	6.40	6.26	6.00	7.11	Fatty, green odor
10	(2E,4E)-2,4-Octadienal	10.0	2.38	16.10	20.34	19.37	14.68	Green, fatty
	Ethers							
15	Dimethyl disulfide	12.0	52.46	67.00	82.63	34.83	45.15	Diffuse, intense onion odor
18	3,4-Dimethyl-thiophene	5000.0	0.09	0.06	0.07	0.05	0.06	Savory roasted onion
21	Dimethyl trisulfide	6.0	683.75	584.10	683.39	373.19	453.70	Fresh onion
24	Diallyl disulfide	30.0	11.94	21.54	27.85	21.42	14.57	Characteristic garlic odor
	Alcohols							
34	(z)-2-Penten-1-ol	720.0	—	—	0.01	—	—	Green, fruity
35	3-Hexen-1-ol	70.0	—	—	0.22	0.14	0.13	Green, leafy
36	Linalool	37.0	—	—	0.32	0.24	0.18	Pleasant floral odor
37	1-Octanol	100.0	0.07	0.23	0.22	0.14	0.13	Green, rose
	Ketones							
48	α-Ionone	10.6	—	1.48	2.64	—	—	Sweet, floral, fruity
49	Geranylacetone	60.0	0.56	0.69	—	—	—	Fresh, green, fruity, rose
50	β-ionone	8.4	16.48	14.37	27.83	12.47	20.78	Flowery
	Esters							
58	*Trans*-3-Hexenyl acetate	870.0	—	0.02	0.04	0.03	0.02	Fruity, green, banana, pear
60	Ethyl myristate	4000.0	0.010	0.012	0.008	0.008	0.005	Sweet waxy violet
61	Hexadecanoic acid, ethyl ester	2000.0	0.13	0.11	0.06	0.11	0.10	Fruity
	Phenols							
67	Eugenol	7.1	2.75	1.51	4.92	3.34	3.58	Sweet, spicy, clove
68	2-Methoxy-4-vinylphenol	12.0	3.04	4.69	3.39	3.09	3.02	Sweet, spicy, clove
	Furans							
70	2-Pentylfuran	5.8	—	—	2.5	—	—	Fruity, green, vegetable

“—”, not found. ^a^ Sequence number of volatile compounds are in agreement with Table 1. ^b^ Thresholds of volatile compounds were obtained from published literature [40,42,43,44]. ^c^ Odor description was obtained from the online database (http://www.thegoodscentscompany.com) (accessed on 20 April 2022).

## Data Availability

All data are contained within the article.

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
