# Peer review of "Moderate Salinity of Nutrient Solution Improved the Nutritional Quality and Flavor of Hydroponic Chinese Chives (Allium tuberosum Rottler)"

_foods, 2023, doi:10.3390/foods12010204_

Round 1

Reviewer 1 Report

1.       Title of manuscript is very simple. Revise it.

2.       Don’t use abbreviation in abstract. Add full form of all abbreviations.

3.       In introduction section, revise this sentence. ‘Vegetables are a fundamental ingredient of human diet, which can provide a variety 32 of nutrients for people, including sugars, minerals, vitamins, dietary fiber and so on’.

4.       Please rephrase the lines 36 to 40 Therefore, improving the quality of vegetable crops can not only enhance their marketability, increase farmers' income, but also enhance their nutritional value and medicinal value, and broaden their utilization.  The formation of vegetable quality is determined by multiple factors, including crop variety, cultivation mode, management measures, environmental conditions and so on…

5.       Similarly, there are a lot of grammatical mistakes please carefully read the whole article to improve the quality of article.

6.       In line 44 and 45 Besides vegetables, Chinese chive is also regarded as medicine, because it has abundant bioactive compounds….. please revise the sentence and mention bioactive components which are present in it.

7.       In introduction section, the term ‘so on’ is frequently used. Use other terms instead.

8.       Author should give rationale of the study in the end of introduction along with proper reasoning of this study.

9.       Please add comprehensive methods so that reader can understand the methodology properly.

10.   Statistical analysis is not clear, please mention which design has been used.

11.   Can you please separate some volatile compounds to shorten the table 1 as it may cause confusion. You can separate on basis of their type or any other that you feel feasible.

12.   Add more practical applications in conclusion section.

13.   Check the references style in the bibliography and uniform.

14.   Check English and grammar errors throughout the manuscript.

15.   Check spelling errors and correct them accordingly.

16.   Discussion section should be improved.

Reviewer 2 Report

The manuscript describes the effect of sodium chloride on nutritional quality and flavor of hydroponic Chinese chives.

1. Need to change title. not clear. include manipulated variables and responding variable. what nutrient solution? too general. sodium chloride? amount? concentration?
2. In abstract, authors mentioned that there are few reports on the improvement of nutrient quality and flavor of hydroponic Chinese 14 chives (Allium tuberosum Rottler) by sodium chloride. so what made your study differ from the available literature? Discuss in introduction.
3. Line 83-84.... few reports mentioned (no citations?). Again, discuss the results of the study.
4. Provide overall flowchart of methodology used in this study, so reader could understand better.
5. Section 2.1 - provide experimental set-up/schematic diagram that reflects with explanation
6. Line 117-118 - slight modifications. what modification that you made? explain. check throughout manuscript. any modification of method need to be explained.
7. Line 132 -134, 167, 173. Equation ? numbers the equation. no references?
8. Line 177 - why you choose DMRT other than post-hoc test? justify and discuss in the manuscript.
9. Fig. 1, 2, 6 too small. make at least half-page.
10. Results and discussion should be combined. discussion based on section. This parts need major amendments.
11. Line 361 - low sodium chloride level of nutrient solution doesn’t inhibit plant growth, and even has a certain growth promoting effect on some plants. Explain the mechanism behind this statement.
12. Line 372 - Chl.a and Chl.T. discuss the difference.
13. Line 408 - safety quality. any criteria for safety quality? how you want to make sure comply with safety quality? 
14. Line 471 - 477 - need comprehensive discussion for PCA. compare with other studies.
15. Line 493 - 494...a certain application basis for the high-quality development of hydroponic Chinese chives industry, and also provide a fresh idea for high-quality cultivation of crops...too general application basis and fresh idea. be specific. 

Round 2

Reviewer 1 Report

Paper can be accepted in present form

Author Response

Thank you for your letter and for the reviewer’s comments concerning our manuscript. Those comments are all valuable and very helpful for revising and improving our paper, as well as the important guiding significance to our research. We have studied comments carefully and have made corrections which we hope meet with approval. For more revision details, please see the latest manuscript.

Reviewer 2 Report

accept in present form 

Author Response

(The authors gave the same response as above.)
